# Isolation and Characterization of an Anti-Osteoporotic Compound from *Melia toosendan* Fructus

**DOI:** 10.3390/pharmaceutics15102454

**Published:** 2023-10-13

**Authors:** Seong Cheol Kim, Dong Ryun Gu, Hyun Yang, Sung-Ju Lee, Jin Ah Ryuk, Hyunil Ha

**Affiliations:** KM Convergence Research Division, Korea Institute of Oriental Medicine (KIOM), Yuseong-daero 1672, Yuseong-gu, Daejeon 34054, Republic of Korea

**Keywords:** *Melia toosendan*, toosendanin, bone resorption, osteoporosis, RANKL

## Abstract

*Melia toosendan* fructus, traditionally employed in traditional Chinese and Korean herbal medicine, exhibits diverse biological properties encompassing anti-tumor, anti-inflammatory, and anti-viral effects. However, its influence on bone metabolism remains largely unexplored. In this study, we investigated the impact of an ethanolic extract of *Melia toosendan* fructus (MTE) on osteoclast differentiation and characterized its principal active constituent in osteoclast differentiation and function, as well as its effects on bone protection. Our findings demonstrate that MTE effectively inhibits the differentiation of osteoclast precursors induced by receptor activator of nuclear factor κB ligand (RANKL). Utilizing a bioassay-guided fractionation approach coupled with UHPLC-MS/MS analysis, we isolated and identified the triterpenoid compound toosendanin (TSN) as the active constituent responsible for MTE’s anti-osteoclastogenic activity. TSN treatment downregulated the expression of nuclear factor of activated T cells c1, a pivotal osteoclastogenic transcription factor, along with molecules implicated in osteoclast-mediated bone resorption, including tumor necrosis factor receptor-associated factor 6, carbonic anhydrase II, integrin beta-3, and cathepsin K. Furthermore, treatment of mature osteoclasts with TSN impaired actin ring formation, acidification, and resorptive function. Consistent with our in vitro findings, TSN administration mitigated trabecular bone loss and reduced serum levels of the bone resorption marker, C-terminal cross-linked telopeptides of type I collagen, in a mouse bone loss model induced by intraperitoneal injections of RANKL. These results suggest that TSN, as the principal active constituent of MTE with inhibitory effects on osteoclastogenesis, exhibits bone-protective properties by suppressing both osteoclast differentiation and function. These findings imply the potential utility of TSN in the treatment of diseases characterized by excessive bone resorption.

## 1. Introduction

Throughout adulthood, bones undergo constant remodeling to maintain their mass and skeletal integrity. This intricate process of bone homeostasis relies on the interaction between two crucial cell types: osteoblasts, responsible for bone formation, and osteoclasts, dedicated to bone resorption. In numerous pathological conditions, an imbalance occurs, leading to excessive bone resorption relative to bone formation, resulting in a net loss of bone. Enhanced osteoclast formation and function have been implicated in a range of bone metabolic disorders, including postmenopausal osteoporosis, bone erosions associated with rheumatoid arthritis, and osteolysis in metastatic cancers [1,2].

Osteoclasts, which are multinucleated differentiated cells, are formed through the commitment and fusion of monocyte/macrophage lineage cells. Osteoclast differentiation is primarily driven by two cytokines: macrophage colony-stimulating factor (M-CSF) and receptor activator of nuclear factor κB ligand (RANKL). These cytokines are produced by mesenchymal-derived cells such as osteoblasts and osteocytes within bone tissue [2,3]. In conjunction with M-CSF, a pivotal factor for the proliferation and survival of osteoclast precursors, RANKL plays a central role in orchestrating the various stages of osteoclast development, encompassing osteoclast commitment, fusion of osteoclast precursors, and activation of mature osteoclasts [4].

The interaction between RANKL and its receptor, RANK, which is present on osteoclast precursors, triggers RANK trimerization and recruits RANK adaptor molecules, tumor necrosis factor receptor-associated factors (TRAFs). Among these TRAFs, TRAF6 plays a pivotal role in RANK-mediated osteoclast differentiation [5]. The RANKL/RANK/TRAF6 axis induces and activates the crucial osteoclastogenic transcription factor NFATc1 [6,7]. Subsequently, NFATc1 promotes the expression of osteoclastogenic genes, such as tartrate-resistant acid phosphatase (TRAP), cathepsin K (CtsK), matrix metalloproteinase 9 (Mmp9), and carbonic anhydrase II (CA II), in collaboration with other osteoclastogenic transcription factors, including c-Fos [7,8,9,10,11].

In addition to osteoclast differentiation, the bone-resorbing activity of mature osteoclasts plays a crucial role in bone remodeling. Mature osteoclasts degrade both the inorganic and organic components of the bone matrix by secreting acids and proteases into a sealed resorption compartment formed between their basal surface and the bone surface. This process relies on the distinct cytoskeletal reorganization of osteoclasts, leading to the formation of a ring-like structure known as the actin ring or sealing zone [12]. Acidification of late endosomes/lysosomes and the trafficking of acidified vesicles to the ruffled border of osteoclasts are critical for acidifying the resorption microenvironment, ultimately resulting in the degradation of the bone mineral [12].

*Melia toosendan* Sieb. et Zucc., a member of the Meliaceae family, is a deciduous tree native to southwest China. Its dried mature fruit, *M. toosendan* fructus, known as Chuan Lian Zi in Traditional Chinese Medicine and Korean Medicine, has a long history of traditional use, characterized by its bitter and warm properties, and association with the Liver and Stomach meridians [13]. Prior research has unveiled a broad spectrum of pharmacological activities of *M. toosendan* fructus, including anti-tumor effects [14,15], anti-inflammatory properties [16], analgesic capabilities [17,18], anti-viral effects [19], and anti-bacterial activity [20]. Furthermore, numerous active phytochemical compounds have been isolated and identified from *M. toosendan* fructus, encompassing triterpenoids, organic acids, lignans, steroids, flavonoids, and volatile oils [13]. Despite the diverse pharmacological activities, the impact *M. toosendan* fructus on bone metabolism has remined largely unknown.

In recent years, there has been a growing interest in exploring the potential of utilizing natural products as an alternative treatment strategy for both mitigating pathological bone loss and promoting the maintenance of bone health [21,22]. In the context of this research, while screening herbal extracts for their inhibitory effects on osteoclast differentiation, we discovered that an ethanolic extract of *M. toosendan* fructus (MTE) exhibited notable inhibitory effects. Subsequently, our investigation extended to identify the principal active constituent responsible for MTE’s impact, followed by a comprehensive characterization of its anti-osteoclastogenic and anti-osteoporotic effects.

## 2. Materials and Methods

### 2.1. Materials

Formic acid was acquired from Samchun (Pyeongtaek, Republic of Korea). All HPLC-grade solvents and fetal bovine serum (FBS) were obtained from Thermo Fisher Scientific (Waltham, MA, USA). Minimum Essential Medium, Alpha Modification (α-MEM), was purchased from Hyclone (Logan, UT, USA). AS-MX phosphate, fast red violet LB salt, and naphthol were procured from Sigma-Aldrich (St. Louis, MO, USA). Antibodies targeting c-Fos and NFATc1 were purchased from Santa Cruz Biotechnology (Santa Cruz, CA, USA). Antibodies targeting TRAF6, c-Src, CAII and HRP-conjugated secondary antibodies (anti-rabbit IgG and anti-mouse IgG) were purchased from Cell Signaling Technology (Danvers, MA, USA). TaqMan Universal Master Mix II and TaqMan probes for the target genes, including c-Fos, NFATc1, Prdm1, Irf8, MafB, Ctsk, Itgβ3, Mmp9, and 18S rRNA, were obtained from Applied Biosystems (Foster City, CA, USA). TSN standard was purchased from ChemFace (Wuhan, China).

### 2.2. Extraction and Fractionation of MTE

To prepare the MTE, 500 g of *M. toosendan* fructus were ground into a powder and subjected to reflux in 3.5 L of 70% ethanol for a duration of 3 h. The resulting solution was subsequently filtered through Whatman no. 3 filter paper and subsequently evaporated at a temperature of 60 °C. Following the evaporation process, the filtrate was concentrated using lyophilization, ultimately yielding the ethanol extract referred to as MTE. The crude extract was dissolved in water and subsequently fractionated using dichloromethane, ethyl acetate, and butanol. An active dichloromethane fraction was further purified using silica gel chromatography. The initial fractionation was carried out using a silica column chromatography (silica gel 60, 0.2–0.5 mm, 40 × 500 mm). The dichloromethane fraction was loaded into the column, and fractions of 200 mL were collected with a gradient formed sequentially of hexane:EtOAc (100:0, 80:20, 60:40, 40:60, 20:80, and 100% EtOAc) as the mobile phase. The most active fraction (fraction 7; 320 mg) was then subjected to further purification by high-performance liquid chromatography (Shimadzu, Tokyo, Japan) using a YMC-Actus Triart C18 column (20 × 250 mm, 5 μm, YMC, Kyoto, Japan) to obtain sub-fractions. The most potent active sub-fraction underwent UHPLC-MS/MS analysis for the characterization of bioactive compounds. All extracts and fractions were stored at −20 ℃ until further use.

### 2.3. UHPLC-MS/MS Analysis

To analyze the constituents within the active fraction of MTE, a Dionex UltiMate 3000 system equipped with a Thermo Q-Exactive mass spectrometer was employed. Chromatographic separation was achieved utilizing an Acquity BEH C18 column (100 × 2.1 mm, 1.7 μm). The mobile phase consisted of water (A) and acetonitrile (B) with 0.1% formic acid, and elution followed a gradient pattern as follows: 0–1 min, 3% B; 1–2 min, 3–15% B; 2–13 min, 15–50% B; 13–20 min, 50–100% B; 20–23 min, 100% B; and 23.5–27.5 min, 3% B, flowing at a rate of 1 mL/min. The Q-Exactive mass spectrometer was equipped with a heated electrospray ionization source and operated in negative mode.

### 2.4. BMM Culture and Cell Viability Assay

BMMs were prepared from mouse bone marrow cells and maintained in α-MEM complete medium containing 10% FBS and M-CSF (60 ng/mL), following a previous protocol [23]. BMMs (2 × 10^4^ cells/well) were seeded in 96-well plates. After 12 h, the culture medium was replaced with fresh complete medium with or without MTE (0–1000 μg/mL) for a 24 h period. Subsequently, the CCK-8 reagent was added, and the absorbance was measured at 450 nm using a standard microplate reader (Molecular Devices, San Jose, CA, USA).

### 2.5. Osteoclast Differentiation Assay

BMMs were incubated with or without MTE (0–100 μg/mL) and its fractions (0–30 ng/mL) for 3 days in the presence of M-CSF (60 ng/mL) and RANKL (50 ng/mL). To evaluate total TRAP activity, cells were incubated with 0.1% Triton X-100 in TRAP buffer including 550 mM sodium tartrate, 0.12 M sodium acetate, and p-nitrophenyl phosphate for 15 min. The reaction was stopped by NaOH (0.1 M) treatment, and the absorbance was measured at 405 nm. After measuring TRAP activity, the cells were stained using TRAP staining buffer, as previously described [23].

### 2.6. Actin Ring and Acridine Oragne Staining

Mature osteoclasts were obtained using a previously described method [24] with a slight modification. Briefly, BMMs were cultured for 2 days in a 10 cm tissue culture dish with M-CSF and RANKL. After detaching the cells using a cell dissociation buffer (Thermo Fisher Scientific), they were transferred to a culture dish coated with a collagen gel. Subsequently, the cells were cultured for an additional 2 days with M-CSF and RANKL to induce the formation of fully mature osteoclasts. The mature osteoclasts were detached using collagenase (Sigma-Aldrich) and seeded in 96-well plates for actin ring and acridine orange staining. After settling for 3 h, the cells were cultured with or without MTE and its fractions in the presence of M-CSF and RANKL for an 18 h period. For actin ring staining, the cells were fixed with 10% formalin, washed with PBS, and F-actin was stained with phalloidin-tetramethylrhodamine B isothiocyanate (Sigma-Aldrich). For acridine orange staining, cells were incubated in α-MEM medium with 5 μg/mL of acridine orange (Sigma-Aldrich) for 15 min, washed, and further incubated in fresh α-MEM medium for 10 min. All fluorescence images were captured using the ImageXpress Micro 4 imaging system (Molecular Devices).

### 2.7. Osteoclast Activity Assay

Mature osteoclasts were obtained, seeded on a Corning Osteo Assay Surface plate (Corning Inc., Corning, NY, USA), and treated with or without its fractions as described above. After culturing for 18 h, the cells were removed using hypochlorite solution, and the plate surface was photographed. ImageJ software (version 1.53) was applied to quantify the resorbed area.

### 2.8. Western Blot Anlysis

BMMs treated as indicated were washed with PBS and lysed using a lysis buffer for protein extraction. The proteins were separated using SDS-PAGE and transferred onto polyvinylidene difluoride membranes. After blocking with 5% skim milk, the membranes were incubated overnight with specific primary antibodies (diluted 1:1000) at 4 °C. After washing three times with a TBS-T buffer, the membranes were probed with corresponding HRP-conjugated secondary antibodies (diluted 1:2000) at room temperature for 1 h. The target bands were detected using SuperSignal^®^ West Pico Chemiluminescent Substrate. Lastly, the chemiluminescent signals were analyzed using a ChemiDoc Imaging System from Bio-Rad (Hercules, CA, USA).

### 2.9. Quantitative Real-Time Polymerase Chain Reaction (PCR)

Total RNA extraction from BMMs and cDNA synthesis were carried out as previously described [23]. The synthesized cDNA was then amplified using TaqMan Universal Master Mix II and TaqMan probes specific to the target genes, employing the ABI 7500 Real-Time PCR Instrument (Applied Biosystems).

### 2.10. Animal Experiment

Female C57BL/6 J mice (7 weeks old) were obtained from SLC Inc. (Shizuoka, Japan) and maintained under specific pathogen-free conditions in accordance with standard laboratory conditions. After one week of acclimatization, healthy mice were randomly assigned to four groups (*n* = 7): Normal group, RANKL-injected group, RANKL + TSN 0.2 mg/kg/day treatment group (TSN-0.2), and RANKL + TSN 1 mg/kg/day treatment group (TSN-1). The mice were intraperitoneally injected with either saline (vehicle) or TSN daily for 6 days. To induce bone loss, RANKL (1.5 mg/kg/day) or PBS was intraperitoneally injected on days 4, 5, and 6. One day after the final RANKL injection, serum and femoral bone samples were collected following a 6 h fasting period. Serum CTX and PINP levels were measured following the manufacturer’s instructions (Immunodiagnostic Systems Ltd., London, UK). Trabecular bone microstructure analysis was performed on the distal end of the femur using micro-CT scan (SkyScan 1276 system, Bruker, Kontich, Belgium). The acquired images were reconstructed and subjected to bone morphometric analysis using SkyScan software (version 1.7.42, Bruker), as previously described [23].

### 2.11. Statistical Analysis

Data are presented as mean ± standard deviation (SD) for in vitro experiments and mean ± standard error of the mean (SEM) for animal experiments. Values were assessed via one-way analysis of variance (ANOVA) with Dunnett’s multiple range test or two-way ANOVA with Sidak’s post hoc test. A value of *p* < 0.05 was regarded as statistically significant.

## 3. Results

### 3.1. Inhibitory Effects of MTE and Its Fractions on Osteoclast Differentiation

We assessed the potential of MTE to influence osteoclast differentiation by evaluating its effect on RANKL-induced differentiation in osteoclast precursors using mouse BMMs. The extent of differentiation was determined by measuring the overall cellular TRAP activity and counting TRAP-positive multinucleated cells, which are recognized indicators of osteoclasts. Our results showed that MTE significantly inhibited osteoclast differentiation at a concentration of 10 μg/mL. Importantly, no cytotoxic effects were observed in BMMs even when MTE concentration reached 100 μg/mL (Figure 1A–D).

To elucidate if specific components within MTE were responsible for this inhibitory action, we proceeded to fractionate MTE using a sequence of organic solvents: dichloromethane, ethyl acetate, and butanol. This yielded distinct fractions and a residual water fraction. Intriguingly, the dichloromethane fraction (10 μg/mL) emerged as the most effective inhibitor of osteoclast differentiation, whereas the other fractions (10 μg/mL) did not show any inhibitory properties (Figure 1B–D). Based on these findings, we have chosen the dichloromethane fraction for subsequent isolation and detailed characterization.

### 3.2. Bioassay-Guided Isolation of Bioactive Compound TSN from MTE

The dichloromethane fraction underwent separation via silica gel column chromatography, producing 11 fractions (fractions 1–11). Notably, fractions 6 and 7 (3–30 ng/mL) showed dose-dependent inhibition of osteoclast differentiation, whereas the other fractions lacked such effects (Figure 2A,B). Fraction 7, exhibiting the most pronounced inhibitory activity, was further partitioned, yielding four sub-fractions (7a–7d). Among these, sub-fraction 7c most effectively inhibited osteoclast differentiation (Figure 2A,B). Figure 3A provides a schematic representation of the entire bioassay-guided fractionation procedure.

To determine the constituents of active sub-fraction 7c, ultrahigh-performance liquid chromatography-tandem mass spectrometry (UHPLC-MS/MS) was utilized, as outlined in Section 2. Through this, TSN, characterized by [M-H]^−^ at *m*/*z* 573.2334, was identified within sub-fraction 7c (Figure 3B). In line with this, the isolated compound TSN hindered osteoclast differentiation at concentrations analogous to those found in sub-fraction 7c (Figure 3C).

### 3.3. Mechanistic Insights into TSN’s Inhibition of Osteoclast Differentiation

TSN’s pronounced inhibitory effect on osteoclast differentiation prompted us to delve into its influence on the mRNA and protein expression levels of key transcription factors and osteoclast-specific genes governing osteoclastogenesis. Notably, TSN treatment at 30 ng/mL curtailed the expression of essential osteoclastogenic transcription factors like c-Fos and NFATc1 during RANKL-mediated osteoclast differentiation, evident at both mRNA and protein tiers (Figure 4A,B). Additionally, the compound downregulated the mRNA expression levels of Mmp9, integrin beta-3 (Itgβ3), and Ctsk (Figure 4B). These genes are characteristically expressed in mature osteoclasts and play pivotal roles in their bone resorptive function [3].

Osteoclastogenesis, under the influence of RANKL, also involves the suppression of negative regulators such as v-maf avian musculoaponeurotic fibrosarcoma oncogene homolog B (MafB) and interferon regulatory factor 8 (Irf8) [25,26]. A previous study showed that the expression of B lymphocyte-induced maturation protein-1 (Blimp1), encoded by the Prdm1 gene, is upregulated by NFATc1. This upregulation promotes osteoclastogenesis by suppressing the negative regulators MafB and Irf8 [27].

Consistent with the attenuation in NFATc1 expression, TSN impeded the expression of Prdm1. Intriguingly, the decrease in MafB and Irf8 expression instigated by RANKL remained largely unaltered by TSN treatment (Figure 4B). This implies that the modulation of these negative regulators might not be central to the mechanism by which TSN affects NFATc1 expression.

Furthermore, our observations disclosed that TSN treatment led to diminished protein expression levels of TRAF6, c-Src, and CA II (Figure 4C), which are pivotal to osteoclast’s bone resorptive function [3,5]. The decline in TRAF6 with TSN exposure was consistent across concentrations ranging from 1 to 30 ng/mL. The downregulation of CA II followed a dose-dependent trajectory, whereas the decrement in c-Src was conspicuous only at elevated TSN concentrations, specifically 30 ng/mL.

### 3.4. Impact of TSN on the Resorptive Function of Mature Osteoclasts

Having confirmed TSN’s suppressive role in osteoclast differentiation, we further explored its influence on the functionality of mature osteoclasts. When mature osteoclasts were cultured on a plate with a bone-mimetic synthetic surface, they formed resorption pits as a result of their acid secretion capabilities. Notably, TSN treatment led to a marked and dose-dependent decrease in such resorption activities (Figure 5A,B).

To elucidate the influence of TSN (1–30 ng/mL) on actin ring architecture, a hallmark of osteoclast functionality, we employed fluorescent phalloidin to stain F-actin within mature osteoclasts. These osteoclasts usually exhibit a distinctive ring-like arrangement of F-actin, known as the actin ring, predominantly around the cell’s periphery. This intricate actin arrangement was notably perturbed upon TSN exposure, and this disturbance followed a dose–response pattern (Figure 5A,C).

To evaluate acidification in mature osteoclasts, we utilized acridine orange staining. Acridine orange accumulates in the acidic compartments of living cells, leading to a shift in fluorescence from green ueutral conditions to red in acidic environments. Acridine orange staining of living osteoclasts demonstrated acidification in normally matured osteoclasts. However, TSN (1–30 ng/mL) exposure led to a dose-dependent reduction in this acidification, as evidenced by diminished red fluorescence (Figure 5A,D).

### 3.5. Protective Effect of TSN on RANKL-Induced Bone Loss

Upon determining the inhibitory potential of TSN on osteoclastogenesis and its anti-resorptive activity, we further assessed its therapeutic efficacy in a mouse model of RANKL-induced bone loss. As documented by Tomimori et al. [28], daily intraperitoneal injections of RANKL for three days in female C57BL/6J mice precipitated a rapid decline in trabecular bone mass, evident a day post the final RANKL administration (Figure 6A). Subsequent micro-CT evaluations indicated pronounced reductions in the trabecular bone mineral density, bone volume, number, and thickness, accompanied by augmented trabecular separation within the femoral bone following RANKL challenge. Notably, treatments with TSN (administered at doses of 0.2 and 1 mg/kg/day) substantially mitigated these bone deteriorative changes (Figure 6B).

To discern the underlying molecular mechanisms accounting for the bone-preservation by TSN, we quantified the serum levels of C-terminal cross-linked telopeptides of type I collagen (CTX, indicative of bone resorption) and procollagen type I N-terminal propeptide (PINP, a marker reflecting bone formation). Mice subjected to RANKL injections exhibited elevated serum concentrations of both CTX and PINP relative to the control cohort. Remarkably, TSN treatments resulted in a marked reduction in the serum levels of CTX and PINP when juxtaposed with the RANKL-only group (Figure 6B). Collectively, these observations underscore the primary role of TSN in curtailing osteoclast-driven bone resorption, conferring its bone-preserving attributes.

## 4. Discussion

In this study, we demonstrated the inhibitory effect of MTE on osteoclast differentiation and identified the active constituent, TSN, primarily responsible for MTE’s impact through a bioassay-guided fractionation approach coupled with UHPLC-MS/MS analysis. Furthermore, we elucidated TSN’s inhibitory impact and the mechanisms underlying osteoclast differentiation and function.

Recent research has shown that TSN possesses diverse beneficial effects in animal experiments, including a broad-spectrum anti-cancer activity [29], anti-obesity effects [30], anti-viral properties [31], and anti-inflammatory and anti-colitis effects [32]. Furthermore, a recent study demonstrated that TSN suppresses osteoclast formation and attenuates osteoporosis in an ovariectomized mouse model [33]. This study has shown that treatment of osteoclast precursor cells with TSN inhibits RANKL-stimulated expression of c-Fos and NFATc1 by suppressing the activation of p38 mitogen-activated protein kinase (MAPK), subsequently inhibiting the expression of osteoclast-specific genes and osteoclast differentiation. Similarly, in our study, TSN was found to suppress RANKL-induced expression of c-Fos and NFATc1. Interestingly, TSN did not affect RANKL-induced down-regulation of negative regulators for NFATc1 and osteoclastogenesis, Irf8 and MafB, particularly at the early stage of osteoclast differentiation. It should be noted that in osteoclast precursors RAW264.7 cells and BMMs, the activation of p38 and JNK MAPKs is required for RANKL-induced MafB downregulation [25].These results suggest that other mechanisms are also involved in TNS’s inhibitory effects on the expression of NFATc1 and osteoclast differentiation.

TRAF6 plays a pivotal role in both osteoclast differentiation and activation. This protein comprises four major domains: the N-terminal RING, zinc fingers, a coiled-coil, and the C-terminal TRAF domains. Notably, the second and third zinc finger domains of TRAF6 are known to play crucial roles in osteoclast differentiation, whereas the RING finger domain is essential for osteoclast activation to resorb bone, albeit not in the differentiation phase [5]. In our study, we observed a reduction in TRAF6 protein expression following TNS treatment, suggesting that the reduction in TRAF6 may contribute to TNS’s impact on osteoclast differentiation.

In addition to TSN’s inhibitory effects on osteoclast differentiation, we also found its inhibitory impact on the resorptive function of mature osteoclasts. The activation of differentiated multinucleated osteoclasts is required for bone resorption [4]. The interaction of integrin αvβ3, highly expressed on osteoclasts, with the bone surface induces actin ring formation and creates a sealed resorption compartment through the formation of a complex with several molecules, including c-Src. Osteoclasts acidify the resorption compartment by secreting protons produced by the enzyme CA II, leading to the decalcification of the inorganic bone matrix [3,12]. Acidification of the resorption compartment also plays a role in the degradation of the bone collagen matrix by activating MMPs and Ctsk enzymes [4].

In our study, treatment of mature osteoclasts with TSN disrupted actin ring structure and inhibited acidification, indicating that these mechanisms play a role in TSN’s anti-resorptive function in mature osteoclasts. Although TSN decreased the protein levels TRAF6, c-Src, and CA II, as well as the mRNA expression of Mmp9, Itgβ3, and Ctsk during osteoclast differentiation, further research is needed to determine whether the regulation of these genes and proteins directly contributes to TNS’s impact on mature osteoclasts.

Consistent with its in vitro inhibition of osteoclast differentiation and function, intraperitoneal administration of TSN (0.2 and 1 mg/kg/day) significantly inhibited trabecular bone loss in mice induced by intraperitoneal injections of RANKL. This RANKL injection model induces rapid trabecular bone loss by increasing osteoclast numbers and stimulating the activation of preexisting osteoclasts [28]. TSN administration suppressed RANKL-induced serum levels of CTX, a bone type I collagen degradation product, indicating that TSN’s bone-protective effect is attributed to its ability to both inhibit osteoclast differentiation and bone resorptive function of osteoclasts. Consequently, our findings suggest the potential of TSN for treating various bone diseases caused by enhanced osteoclast differentiation and function. Notably, our findings on TSN’s anti-osteoporotic activity, coupled with its broad-spectrum anti-cancer activity [29], warrant consideration for the treatment of osteolytic bone metastasis.

It is worth noting that TSN has been previously reported to induce hepatotoxicity. Liang et al. [34] reported acute liver injury in mice following intraperitoneal administration of TSN at doses of 10 and 20 mg/kg. Furthermore, oral administration of TSN at a dosage of 80 mg/kg/day for nine consecutive days resulted in notable body weight loss and severe liver injury characterized by hepatocyte necrosis, while a lower dose of 40 mg/kg of TSN exhibited normal liver histology [35]. Notably, in the same study, mice exposed to TSN (80 mg/kg/day) for 21 days displayed signs of hepatic adaptation to TSN, eident through the recovery of hepatic histology, body weight, and serum levels of liver injury markers. Despite its hepatotoxic effects at doses exceeding 10 mg/kg, it is worth highlighting that many of TSN’s beneficial properties, including its anti-cancer [36], anti-obesity [32], anti-viral [31], and anti-colitis effects [32], have been observed at significantly lower doses ranging from 0.1 to 1 mg/kg/day, without causing toxicity in major organs, including the liver. Furthermore, a study conducted by Jin et al. [37] demonstrated that pretreatment with quercetin, a widely recognized flavonoid known for its antioxidant properties, effectively reversed acute liver injury induced by a single injection of TSN at a dose of 10 mg/kg in mice. This discovery suggests a potential strategy for mitigating TSN-induced hepatotoxicity.

Collectively, these findings underscore the imperative for further research and clinical studies to validate both the efficacy and safety of TSN as an anti-resorptive agent for the treatment of various bone-destructive diseases. Several uncertainties remain, such as the stability of TSN, specific mechanisms and impact on hormone levels underlying its in vivo effects. Conducting further research to address these remaining questions could significantly contribute to advancing TSN’s potential as a therapeutic agent or dietary supplement for a variety of pathological bone conditions.

## 5. Conclusions

This study employed a bioassay-guided fractionation and chemical investigation to explore the anti-osteoclastogenic effects of MTE. As a result, we have successfully identified the active compound, TSN. Our research provided a comprehensive understanding of TSN’s inhibitory effects on osteoclast differentiation and function in vitro. Additionally, we have demonstrated its effectiveness in reducing bone resorption in an animal experiment. These findings hold promise for the potential therapeutic applications of both MTE and TSN in addressing conditions characterized by excessive bone resorption, offering new possibilities for advancing the treatment of bone diseases. Furthermore, these findings serve as the foundational basis for additional research into the anti-osteoporotic activity of TSN synthetic compounds and other biological activities of TSN. Through this, they suggest the potential development of TSN as a pharmaceutical agent.

## Figures and Tables

**Figure 1 pharmaceutics-15-02454-f001:**
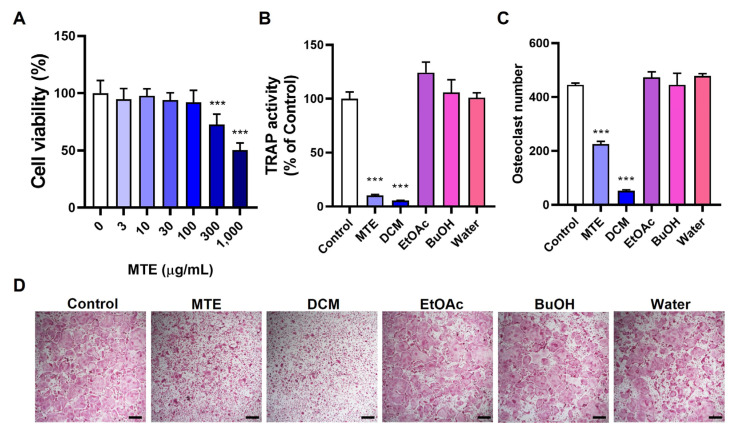
Effects of MTE and its fractions on osteoclast differentiation. (**A**) Cell viability of BMMs treated with MTE ranging from 0 to 1000 μg/mL for 24 h, assessed via the CCK-8 assay. (**B**–**D**) BMMs were incubated with MTE and its fractions at 10 μg/mL in the presence of M-CSF (60 ng/mL) and RANKL (50 ng/mL) for 3 days: (**B**) Total cellular TRAP activity, (**C**) Enumeration of osteoclasts, and (**D**) Representative TRAP-stained cell images (scale bar, 100 µm). *** *p* < 0.001 versus control. Abbreviations: DCM, dichloromethane; EtOAc, ethyl acetate; BuOH, butanol.

**Figure 2 pharmaceutics-15-02454-f002:**
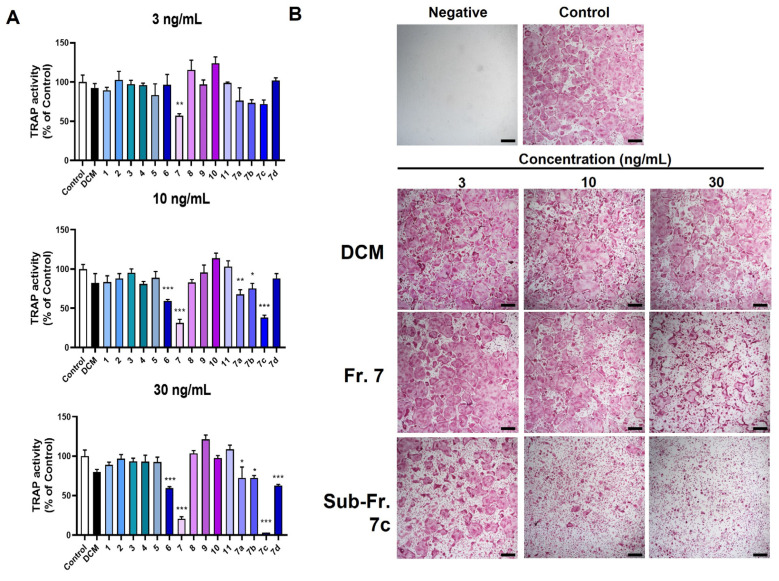
Effects of dichloromethane-derived fractions of MTE on osteoclast differentiation. BMMs were exposed to the dichloromethane fraction of MTE, including its derived fractions (fractions 1–11 and sub-fractions 7a–7d), at concentrations of 3, 10, and 30 ng/mL alongside RANKL for 3 days. (**A**) Evaluation of total cellular TRAP activity. (**B**) Representative images of TRAP-stained cells (scale bar, 100 µm). * *p* < 0.05, ** *p* < 0.01, *** *p* < 0.001 versus control.

**Figure 3 pharmaceutics-15-02454-f003:**
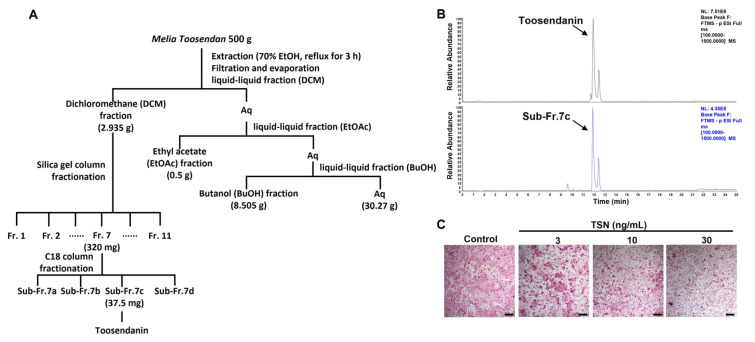
Bioassay-guided identification of the anti-osteoclastogenic compound TSN from MTE. (**A**) Schematic representation of MTE fractionation. (**B**) Detection of TNS standard (upper panel) and TSN in sub-fraction 7c (bottom panel) via UHPLC-MS/MS anlysis. (**C**) Influence of TSN (3, 10, and 30 ng/mL) on RANKL-induced osteoclast differentiation in BMMs (scale bar, 100 µm).

**Figure 4 pharmaceutics-15-02454-f004:**
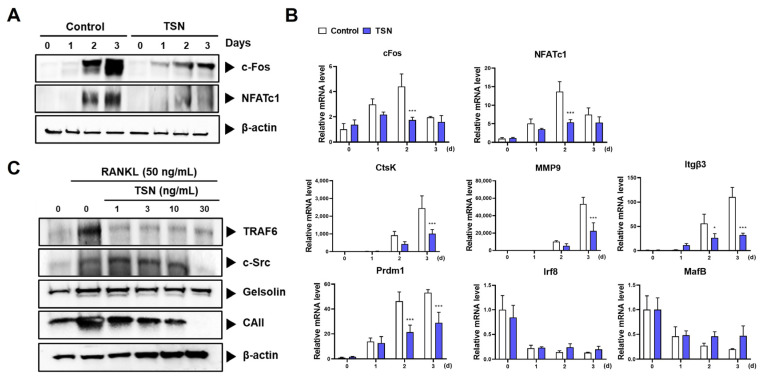
TSN-mediated modulation of osteoclastogenic markers. BMMs were cultured in the presence or absence of RANKL (50 ng/mL) and TSN (30 ng/mL) for 3 days. (**A**) Protein expression of c-Fos and NFATc1 during osteoclast differentiation with or without TSN. (**B**) mRNA expression of selected genes during osteoclast differentiation in the presence or absence of TSN. (**C**) Protein expression of TRAF6, c-Src, and CAII in BMMs exposed to varying concentrations of TSN (1–30 ng/mL) in the presence of RANKL for 3 days. * *p* < 0.05, *** *p* < 0.001 versus control not treated with TSN.

**Figure 5 pharmaceutics-15-02454-f005:**
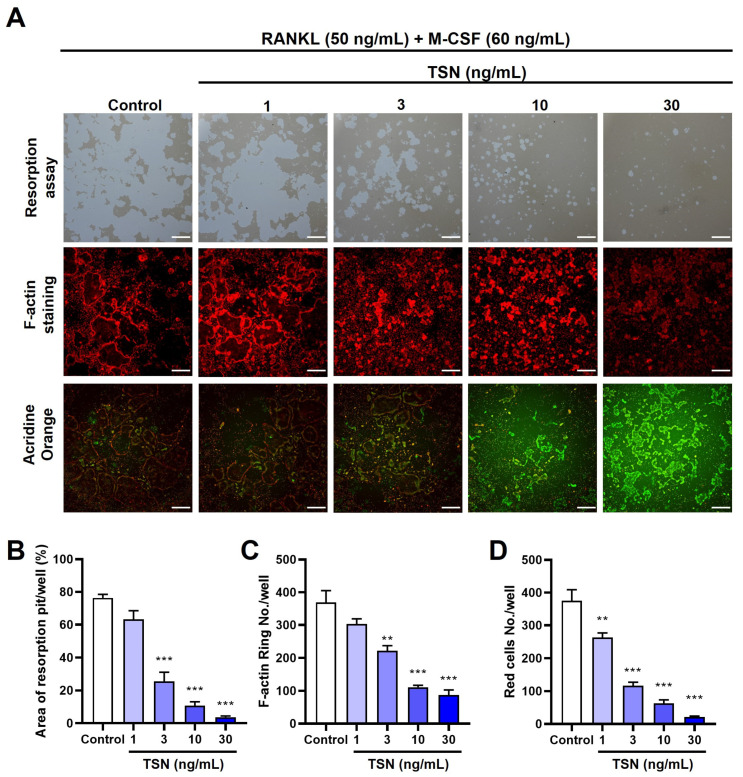
TSN suppresses bone resorption activity, actin ring formation, and acidification in mature osteoclasts. Mature osteoclasts were treated with TSN (1–30 ng/mL) for 18 h, using either Osteo Assay Surface plates or standard tissue culture plates. (**A**) Representative images displaying resorbed surface on Osteo Assay Surface plates (upper panel, scale bar, 200 µm), F-actin staining (middle panel, scale bar, 200 µm), and acridine orange staining (lower panel, scale bar, 200 µm). Quantitative analyses of the resorbed area (**B**), actin rings (**C**), and acidification (**D**) are shown. ** *p* < 0.01, *** *p* < 0.001 versus control without TSN treatment.

**Figure 6 pharmaceutics-15-02454-f006:**
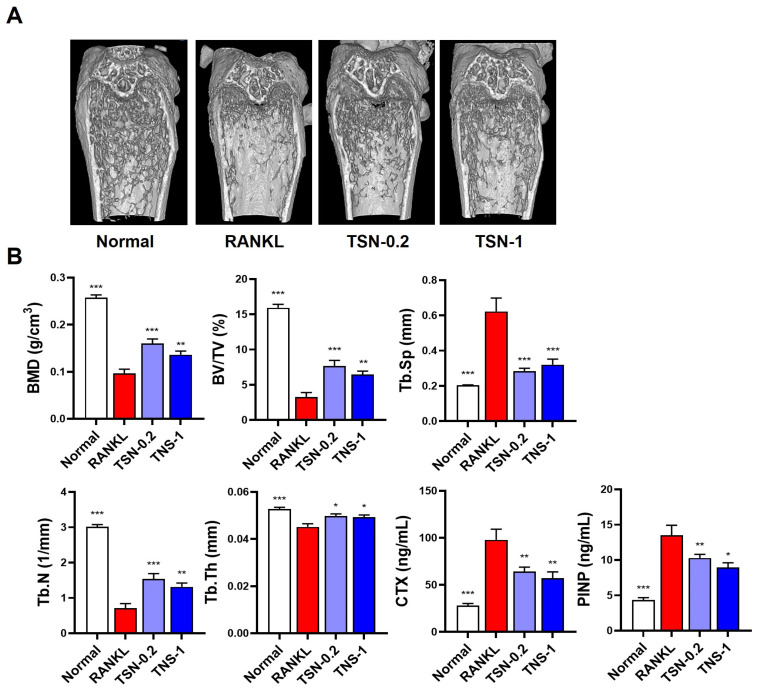
Preventive effect of TSN on RANKL-induced bone degradation in mice. Prior to RANKL administration, mice received pretreatments of either vehicle (saline), TSN at a dose of 0.2 mg/kg/day (TNS-0.2), or 1 mg/kg/day (TNS-1) over a span of 3 days. This was followed by intraperitoneal RANKL injections at 1.5 mg/kg/day for three successive days (**A**) Three-dimensional visualization of the distal femoral bone by micro-CT. (**B**) Bone morphometric analysis of the femoral metaphysis, as well as serum CTX and PINP levels. Abbreviations: BMD, trabecular bone mineral density; BV/TV, bone volume to total volume; Tb. Sp, trabecular separation; Tb. N, trabecular number; Tb. Th, trabecular thickness. * *p* < 0.05, ** *p* < 0.01, *** *p* < 0.001 versus the RANKL-only group.

## Data Availability

All data of this study are provided within the manuscript.

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
