# Peer review of "Isolation and Characterization of an Anti-Osteoporotic Compound from Melia toosendan Fructus"

_pharmaceutics, 2023, doi:10.3390/pharmaceutics15102454_

Round 1

Reviewer 1 Report

The manuscript is well written and organized. The results and discussion are clear. The authors could add in the discussion the limitations of the applied methodology. In material and methods section the concentration of MTE used on each assay should be indicated. Also,  the primary and secondary antibodies used  must be detailed.

Reviewer 2 Report

The manuscript entitled "Isolation and characterization of an anti-osteoporotic compound from Melia toosendan fructus" aims to investigate the impact of the ethanolic extract of Melia toosendan fructus (MTE) on osteoclast differentiation and characterized its principal active constituent in osteoclast differentiation and function, as well as its effects on bone protection.

Several remarks

Make clear all figures. Increase the font.

Add the cytotoxic effect of plant extract.

The conclusions match the obtained results.

The Material and Method chapter is sufficiently detailed.

The novelty of the study is the identification of active compound named TSN and its inhibitory effects on osteoclast differentiation and function in vitro.

Author Response

Thank you for your comment. We have made modifications to clarify certain aspects of the figures and increased the font size. Additionally, while we previously reported the cell viability related to the cytotoxic effect of ethanolic extract of Melia toosendan fructus (MTE) in Figure 1A, we only showed cell viability at concentrations below 100 µg/mL. To supplement this, in relation to the cytotoxicity of MTE extract, we confirmed cell toxicity up to 1,000 µg/mL by using CCK-8 assay. The results showed that MTE extract started affecting cell viability at approximately 300 µg/mL, with no observable cytotoxicity at concentrations as low as 100 µg/mL. The updated data is as follows, and we have amended the manuscript accordingly.

Reviewer 3 Report

This manuscript is interesting for pharmaceutics community and could be accepted for publication. The topic is up to date and actual. Melia toosendan fructus, traditionally employed in traditional Chinese and Korean herbal medicine, exhibits diverse biological properties encompassing anti-tumor, anti-inflammatory, and anti-viral effects. However, its influence on bone metabolism remains largely unexplored. In this study, authors investigated the impact of an ethanolic extract of Melia toosendan fructus (MTE) on osteoclast differentiation and characterized its principal active constituent in osteoclast differentiation and function, as well as its effects on bone protection. Author’s findings demonstrate that MTE effectively inhibits the differentiation of osteoclast precursors induced by receptor activator of nuclear factor κB ligand (RANKL). Utilizing a bioassay-guided fractionation approach coupled with UHPLC-MS/MS analysis, authors isolated and identified the triterpenoid compound toosendanin (TSN) as the active constituent responsible for MTE's anti-osteoclastogenic activity. TSN treatment downregulated the expression of nuclear factor of activated T cells c1, a pivotal osteoclastogenic transcription factor, along with molecules implicated in osteoclast-mediated bone resorption, including tumor necrosis factor receptor-associated factor 6, carbonic anhydrase II, integrin beta-3, and cathepsin K. Furthermore, treatment of mature osteoclasts with TSN impaired actin ring formation, acidification, and resorptive function. Consistent with in vitro findings, TSN administration mitigated trabecular bone loss and reduced serum levels of the bone resorption marker, C-terminal cross-linked telopeptides of type I collagen, in a mouse bone loss model induced by intraperitoneal injections of RANKL. These results suggest that TSN, as the principal active constituent of MTE with inhibitory effects on osteoclastogenesis, exhibits bone-protective properties by suppressing both osteoclast differentiation and function. These findings imply the potential utility of TSN in the treatment of diseases characterized by excessive bone resorption. I have only two suggestions for minor revision. First, it would be a good idea to cite following relevant references about more common (including synthetic) compounds featuring anti-tumor, anti-inflammatory, and anti-viral effects in introduction: J. Am. Chem. Soc. 1966, 88, 3888; Journal of Molecular Structure 2016 1111, 142-150; Molecules 2020, 25, 5776; ACS omega 2017 2 (4), 1380-1391; Inorganic Chemistry 2022 61 (4), 2105-2118; Archives of Pharmacal Research 1996, 19, 228–230; J. Am. Chem. Soc. 2022, 144, 314–330. Second, some more detailed perspectives about the future research could be formulated in conclusions.
